# The Primary Cilium and Neuronal Migration

**DOI:** 10.3390/cells11213384

**Published:** 2022-10-26

**Authors:** Julie Stoufflet, Isabelle Caillé

**Affiliations:** 1Laboratory of Molecular Regulation of Neurogenesis, GIGA-Stem Cells and GIGA-Neurosciences, Interdisciplinary Cluster for Applied Genoproteomics (GIGA-R), University of Liège, CHU Sart Tilman, 4000 Liège, Belgium; 2Inserm U1130, Institut de Biologie Paris Seine (IBPS), Neuroscience Paris Seine (NPS), Sorbonne University, CNRS UMR8246, 75005 Paris, France; 3University of Paris Cité, 75020 Paris, France

**Keywords:** primary cilium, neuronal migration

## Abstract

The primary cilium (PC) is a microtubule-based tiny sensory organelle emanating from the centrosome and protruding from the surface of most eukaryotic cells, including neurons. The extremely severe phenotypes of ciliopathies have suggested their paramount importance for multiple developmental events, including brain formation. Neuronal migration is an essential step of neural development, with all neurons traveling from their site of birth to their site of integration. Neurons perform a unique type of cellular migration called cyclic saltatory migration, where their soma periodically jumps along with the stereotyped movement of their centrosome. We will review here how the role of the PC on cell motility was first described in non-neuronal cells as a guide pointing to the direction of migration. We will see then how these findings are extended to neuronal migration. In neurons, the PC appears to regulate the rhythm of cyclic saltatory neuronal migration in multiple systems. Finally, we will review recent findings starting to elucidate how extracellular cues sensed by the PC could be intracellularly transduced to regulate the machinery of neuronal migration. The PC of migrating neurons was unexpectedly discovered to display a rhythmic extracellular emergence during each cycle of migration, with this transient exposure to the external environment associated with periodic transduction of cyclic adenosine monophosphate (cAMP) signaling at the centrosome. The PC in migrating neurons thus uniquely appears as a beat maker, regulating the tempo of cyclic saltatory migration.

## 1. Introduction

The primary cilium (PC) is a small rod-shaped organelle emanating from the cell surface of almost all eukaryotic cells, including neurons. The PC is templated by the mother centriole, which is then called the basal body. The PC is non-motile and composed of an axoneme of nine microtubule doublets surrounded by a ciliary membrane. The PC length is generally between 0.5 and 10 µm and its width is between 0.2 and 0.3 µm [1]. The PC is separated from the rest of the cell by the transition zone that filters what enters or exits the ciliary cytoplasm (cilioplasm). The ciliary gate is composed of the transition zone and the transition fibers. The transition fibers are necessary to recruit the IFTs (intraflagellar transport proteins), which are crucial to building and maintaining the PC [2] (Figure 1A), since they are linked to kinesin or dynein molecular motors, which transport proteins along the axoneme.

The PC was initially discovered in the 19th century [3]. Its formation was first described by Sorokin in 1962 by electron microscopy (EM) analysis of fibroblasts and smooth muscle cells [4,5]. He observed that a ciliary vesicle can be attached to the centrioles with the PC growing inside the vesicle, which then fuses with the cell membrane. This allows the PC to continue growing outside of the cell (Figure 1B).

For a long time, the role of the PC remained obscure, and three main hypotheses were proposed to explain the role of this neglected organelle. The first hypothesis was that the PC was an evolutionary remnant organelle with no real function. The second one was that the PC was involved in sequestering the centriole to avoid mitosis entry. The third, and now proven to be true, hypothesis was that the PC is a key cellular sensory organelle [6,7].

Its functional importance is clearly demonstrated by the extremely severe phenotypes of ciliopathies, genetic diseases affecting PC production, maintenance and/or function, which display a wide range of symptoms including brain malformations, retinal dystrophy, cystic kidney disease, liver fibrosis, and skeletal abnormalities [8]. Given the brain malformations present in ciliopathies, over the years, numerous studies have taken interest in the role of the PC in the different steps of nervous system development and particularly in cortical development [9]. These steps include neurogenesis, migration of the newly-born neurons, differentiation, axonal pathfinding, dendritogenesis, synaptogenesis, as well as maintenance and plasticity of the neural circuits. However, only a few studies tackled the role of the PC in migrating neurons [10,11,12,13,14,15,16,17] (see Table 1). Those are the studies that will be detailed and compared in this review.

The PC is a sensory organelle that can sense chemosensory, osmosensory, and mechanosensory environments [18,19]. The most spotlighted ciliary signaling pathway is the Sonic Hedgehog pathway. Patched, Shh receptor, and Smoothened, its effector, can be localized in the PC. Upon binding of Shh to Patched, the repression on Smoothened is relieved and the pathway is activated [20].

Apart from Patched and Smoothened, a large number of membrane receptors display ciliary localization. This is the case for numerous G protein-coupled receptors (GPCRs) [21]. Some calcium channels are also localized in the PC, such as the Transient Receptor Potential (TRP) channels [22]. The ciliary gathering of receptors can lead to extremely high concentrations of second messengers like cAMP and calcium in the cilioplasm. Ciliary-targeted cAMP biosensors revealed the accumulation of cAMP in the PC subsequent to GɑS coupled GPCR activation [23,24,25], reaching a concentration as high as five times more elevated than the non-ciliary cytoplasm. Similarly, FRET-based calcium biosensors showed a high ciliary calcium concentration downstream of ciliary channels [26]. Given its tiny volume and possibility to accumulate membrane receptors, the PC thus appears as a very efficient second messenger concentrator in response to external stimuli.

In this review, we will highlight the known roles of the PC during cell migration, mainly focusing on neuronal migration, and we will discuss by which intracellular mechanisms the PC might regulate neuronal migration.

## 2. The PC as a Migration Guide in Non-Neuronal Cells

The first indication that the PC could play a directional role in migration arose from the seminal observation that, whereas it is randomly oriented in confluent cultures of 3T3 fibroblasts, it is predominantly oriented parallel to the substrate towards the direction of migration in sub-confluent migrating mouse embryonic fibroblasts [27]. Later on, in a wound assay using rat embryonic fibroblasts, the PC orientation was shown to occur prior to the initiation of migration [28], further suggesting that it could be a cause rather than a consequence of the movement direction. The orientation of the PC in the direction of migration was also shown in migrating mouse lung fibroblasts [29] and smooth muscle cells [30].

The Christensen laboratory attempted to uncover how the PC could act on the directionality of PDGFɑ-induced chemotactic migration of mouse embryonic fibroblasts [31,32,33,34]. This series of papers showed that the PC of serum-starved fibroblasts harbors PDGF (platelet derived growth factor) receptors and that PDGFRɑ ciliary signal transduction activates the PI3K (phosphoinositide 3-kinase)–AKT (serine/threonine protein kinase) and MEK1/2 (MAP/ERK kinase1)–ERK1/2 (Extracellular signal-regulated kinase)–RSK (Ribosomal s6 kinase) pathways to reorganize the cytoskeleton, which could participate in the control of the direction of migration. They describe the activation of NHE1 (Na+/H+ exchanger 1) in vesicles targeted to the leading edge of migrating neuroblasts, which would create a subcellular communication from the PC to the forward protrusion and facilitate guidance [31].

The PC also has a directional role in endothelial cell migration. In aortic endothelial cells, the PC was shown to act on actin assembly and focal adhesion formation, with defects in directional migration in ciliary mutants [35]. Moreover, in endothelial cells of the cornea, the PC is necessary for proper endothelial cell morphogenesis during development and repair, with ciliary mutants displaying mislocalization of junctional markers and accumulation of cytoplasmic acetylated tubulin [36].

The PC is also involved in the migration of mesenchymal stem cells to bone remodeling sites, which is crucial for bone homeostasis. In [37], the authors show that TGF-β induces human mesenchymal stem cell recruitment. TGF-β receptors are localized in the PC and the PC is necessary for the chemotactic response of mesenchymal stem cells. Osteospontin, an adhesive molecule, as well as fibronectin, were subsequently shown to be additional PC-dependent chemoattractants for mesenchymal stem cells [38]. The PC-dependent osteospontin pathway involves the actin pathway via CD44 and CDC42.

Together, these studies provide evidence that the PC can play the role of a migration guide in fibroblasts, endothelial cells, and mesenchymal stem cells, mostly by acting on cytoskeleton regulating pathways.

## 3. The PC: A Guide and Beat Maker in Migrating Neurons

Neuronal migration is essential to the proper formation of neural circuits since all neurons must migrate from their site of birth to their site of integration. Defective migration leads to severe brain malformations like lissencephaly or cortical heterotopia and might participate in psychiatric disorders [39].

Maybe because of their highly polarized morphology and their huge nucleus/cytoplasm size ratio, neurons migrate very differently from other cells, suggesting a possible unique role for their PC. Rather than sliding like fibroblasts, neurons intermittently jump by performing the so-called cyclic saltatory migration (Figure 2A). In this process, the nucleus and centrosome move forward in a “two-stroke” cycle, with the centrosome moving first within a swelling in the leading process and the nucleus following subsequently [40,41]. As an example, a typical migrating neuron in the postnatal Rostral Migratory Stream performs the full cycle within 20 min on average, with the nucleokinesis phase, i.e., the real movement phase, lasting for less than 5 min [14].

As mentioned above, the phenotypes of ciliopathies illustrate its paramount importance to nervous system development. Some cases of Joubert syndrome and Meckel–Grubert syndrome display cortical heterotopia, which are considered anatomical hallmarks of migration defects [42,43,44,45], thus suggesting the importance of the PC for neuronal migration in the cortex. This was indeed confirmed by a large screen from the Anton lab, where mouse in utero electroporation of a shRNA library of 30 known ciliopathy genes revealed cortical migration defects for as many as 17 of them, such as BBS1 (Bardet–Biedl syndrome protein 1) and IFT80 (Intraflagellar transport 80) (see Table 1) [10].

In terms of orientation, two types of migration are defined during corticogenesis (Figure 2B). Pyramidal excitatory neurons are born locally in the cortical ventricular zone and migrate short distances radially (relative to the surface of the brain) to the cortical plate, along the basal process of radial glial cells [46]. As opposed to this, inhibitory GABAergic interneurons are born far away in subpallial ganglionic eminences and migrate long distances tangentially (also relative to the surface of the brain) before reorienting radially to integrate the cortical plate [47]. Both radially and tangentially migrating neurons perform similar cyclic saltatory migration, but the two types of migration present obvious morphological and molecular differences, so that the PC function in both might bear similarities as well as differences [40,41,48].

### 3.1. Tangential Migration in the Cortex

In the embryonic cortex, the role of the PC during interneuron tangential migration from the ganglionic eminence to the cortical plate was studied by two different teams [11,12]. They studied cortical interneuron migration after complete ablation of the PC [12] or deletion of Arl13b [11], a small ciliary GTPase. The deletion of Arl113b does not completely ablate the PC; however, the PC loses its ability to respond to extracellular signals [49].

In [11], Arl13b was depleted in interneurons by crossing Arl13b^lox/lox^ mice with Dlx5,6:CRE mice (Dlx5,6: distal-less homeobox 5 and 6; CRE: cre recombinase). This led to an abnormal placement of interneurons with a reduction of the distance traveled at a given time as compared to controls. Moreover, mutant interneurons displayed an abnormal morphology with more branches than controls. To further understand the migration defects of mutated neurons, they were live-imaged in organotypic slices. This revealed globally altered migration streams in the cortex. Analysis of the migration dynamics of individual neurons further showed a reduced frequency of nucleokinesis along with a globally reduced speed of migration. During pauses, mutated neurons also showed an indecisive leading process, as if probing in vain their environment. Upon radial reorientation to their final destination in the cortex, mutated interneurons were less prone to orientate their leading processes towards the cortical plate than controls. Together, these data suggest that the PC can play a role in regulating the rhythm of interneuron migration as well as their directionality in terms of reorientation from tangential to radial migration.

The authors also analyzed the migratory behavior of neurons in microfluidic devices, in which interneurons were exposed to extrinsic guidance cues released by dorsal cortical cells present in the opposing channel. Interestingly, the ciliary mutated interneurons displayed a reduced distance of migration after 72 h of culture. The authors interpreted this phenotype as a defect in the ability to respond to extracellular cues, but we suggest that this might also reflect a slowed-down migration. In an attempt to identify which guidance cues could be involved, the authors showed the ciliary localization of multiple receptors for cues potentially important for neuronal migration, including BDNF (brain-derived neurotrophic factor) receptor TrkB (Tropomyosin receptor kinase B), GDNF (glial cell line-derived neurotrophic factor) receptor GFRα-1, SDF-1 (Stromal cell-derived factor 1) receptors, CXCR4 (CXC motif chemokine receptor 4) and CXCR7 (CXC motif chemokine receptor 7), NRG-1 (neuregulin 1) receptor, ErbB4 (Receptor tyrosine-protein kinase 4), serotonin receptor 6, Slit receptors Robo1 and 2, and HGF/SF (hepatocyte growth factor/scatter factor) receptor c-MET. Most receptors displayed a defective ciliary localization in the Arl13B-mutated PC.

The study by the Métin lab [12] is in line with the former study, further showing the double role of the PC as a guide for reorientation and as a regulator of the rhythm of migration. The authors performed PC genetic deletion by crossing Nkx2.1 Cre mice with Kif3a^lox/lox^ mice. Kif3a (Kinesin Family Member 3A) is a subunit of the kinesin motor, whose deletion prevents PC assembly. Mutated interneurons displayed abnormal distribution with accumulation in the cortical streams. Live-imaging in slices showed their reduced speed mostly due to longer pauses and hence less frequent nucleokinesis as above. Moreover, the directionality of interneurons was affected and, in particular, their reorientation towards the cortical plate. Very similar to the previous study, these data suggest that the PC is involved in the regulation of the rhythmicity of migration as well as in the control of the switch of migration from tangential to radial. To better understand the directionality problem and differently from the previous study, the authors focused on the Shh pathway. Pharmacologic activation of Shh in PC-ablated migrating neurons rescued the defective directionality but not the speed of migration. This suggests that Shh might participate in the reorientation of migrating interneurons without regulating the rhythmicity of their migration.

These two parallel studies thus very convincingly showed the importance of the PC for tangential migration in the embryonic cortex and paved the way for new research avenues.

### 3.2. Tangential Migration in the Rostral Migratory Stream

The rostral migratory stream is the main avenue for tangentially migrating neurons in the postnatal and adult mouse brain [50]. It extends from the ventricular/subventricular zone to the olfactory bulb (Figure 2C). A massive number of neuroblasts are constantly produced in the ventricular/subventricular zone and perform cyclic saltatory migration over a long distance along the rostral migratory stream before reorienting radially to integrate the olfactory bulb. Even though the rostral migratory stream is maintained throughout life [51], some obvious but often underestimated morphological differences exist between postnatal and adult rostral migratory streams. The size of the rostral migratory stream massively decreases with age (along with the quantity of migrating neuroblasts produced in the ventricular/subventricular zone). In the adult, neuroblasts migrate as chains ensheathed by astrocytic tubes, which form at around 3 weeks of age [52] and are thus absent in neonates. Similarly, blood vessels, oriented parallel to the rostral migratory stream along the rostro-caudal axis and important regulators of rostral migratory stream migration, form around the third week of age [53]. At this age, the lateral ventricle extending along the rostral migratory stream also closes [52]. It is thus reasonable to think that adult and postnatal rostral migratory stream neurons are under the influence of different extracellular contacts and cues and thus that the PC might respond to different stimuli.

The first study about the role of the PC in the adult and neonate rostral migratory stream was [13]. The authors stereotaxically injected a lentivirus expressing a dominant negative form of Kif3a or a short hairpin RNA against Ift88 into the ventricular/subventricular zone, preventing, respectively, PC formation or maintenance. Analysis on fixed tissue seven days after infection showed that mutated neuroblasts from the two genotypes were severely delayed in their migration, with an accumulation in the rostral migratory stream. Mutated neurons also displayed an orientation defect with an increase in their backward orientation. These data suggest an important role for the PC in adult rostral migratory stream migration. To further study the dynamic of migration, the authors switched to a traditional ex vivo model [54], in which transfected neonate (P1, P2: Postnatal days 1 and 2) ventricular/subventricular zone explants: were allowed to migrate in Matrigel. Live-imaging revealed a decreased migration speed with longer resting phases and decreased nucleokinesis speed. Mutated neuroblasts also displayed more reversal turns. In cells from GFP::Cent2 mice, where the tagged centrosome can be imaged, the authors also observed a decreased speed of the centrosome during centrokinesis as well as its increased backward movement. These data suggest an important role of the PC in the regulation of migration speed, with an influence over both the nucleus and the centrosome dynamic.

Less than a year later, our paper reinforced and detailed this idea [14]. We similarly analyzed the dynamics of migration of PC-ablated rostral migratory stream neurons, but this time in acute slices of the postnatal rostral migratory stream. To this extent, we electroporated Kif3a^lox/lox^ or Rpgrip1L^lox/lox^ (Retinitis Pigmentosa GTPase Regulator Interacting Protein 1 Like, a protein of the transition zone whose deletion leads to PC-ablation in our system) P2 mice with a plasmid encoding CRE-GFP (mutants) or GFP (controls) and live-imaged rostral migratory stream neurons between P7 and P10. This revealed a slowed-down migration with decreased nucleokinesis frequency and an increased percentage of pausing time. In addition, similar to Matsumoto et al. [13], we observed more backward-oriented neurons (unpublished data). Co-electroporation of a Centrin-RFP expressing plasmid with a plasmid encoding CRE-GFP or GFP alone in the same mice revealed a reduced maximal distance between centrosome and nucleus during centrokinesis, as well as an increase in the frequency of inefficient centrokinesis, i.e., centrokinesis not followed by a nucleokinesis. Even though the system used for live-imaging was different from [13] (organotypic slice versus ex vivo Matrigel explants), the results are very similar, suggesting an important role of the PC in the regulation of the rhythm of migration through centrosome-nucleus coupling. Neither of these two papers really addressed a guidance role for the PC except for the increased backward orientation of the mutated cells, which may be a defect of polarity rather than guidance *per se*. It is also to be noted that in the Matsumoto article [13] as well as in our analysis (unpublished data), the delayed mutated neuroblasts ultimately integrate the different olfactory bulb layers, suggesting that they do not display a major problem of reorientation from tangential to radial migration, different from what was described for embryonic cortical interneurons [11,12]. Finally, neither paper examined the migration dynamics of adult rostral migratory stream neuroblasts using live-imaging, so we cannot fully conclude about the effect of PC ablation in adults.

### 3.3. Radial Migration in the Cortex

Whereas the role of the PC in tangential migration was clearly demonstrated in 2012 [10,11,12], it took surprisingly more time to prove its role in radial migration. This might be due to seemingly negative results of initial studies.

In Arellano et al. [55], the authors very elegantly described the development and maturation of the PC in mouse neocortical neurons by immunolabeling of the adenylate cyclase 3 as a PC marker as well as EM analysis. They, however, considered that, as the centrioles are a core component of the centrosome necessary for migration [56], it should be expected that PC extension would occur only after cells have completed migration. In other words, they could not imagine that the moving centrosome of migrating neurons could act as a basal body anchored at the basis of the PC. This view is contradicted by the articles described above [11,12,13,14], but the question of how to reconcile the movement of the centrosome with its role as a basal body is an essential one, for which some still incomplete answers will be given in part 4.

In addition, the initial paper from the Anton lab [11], reporting major migration defects for Arl13B-mutated tangentially migrating interneurons (see above), did not notice major defects in the positioning of mutated pyramidal neurons at E14, E16 (E: embryonic day) or P0. They thus concluded that Arl13B does not play a major role in radial migration, which could imply that the PC is not important for radial migration.

However, a subsequent article from the same lab [10] proved that this is not the case. As mentioned above, the authors found that the knockdown of as many as 17 ciliopathy genes resulted in delayed radial migration as analyzed by mispositioning of mutated neurons (see Table 1). A similar result was observed in another study for an additional ciliary gene, IFT172 [57]. Aberrant positioning of radially migrating neurons was also observed after disruption of autophagy mediated ciliogenesis through down regulation of ATG5 (autophagy related 5) [16].

It is to be noted that, to our knowledge, live-imaging of all these ciliary mutated radially migrating neurons has not been performed, so that it is not possible to fully understand the role of the PC. However, it can be safely concluded that the PC is important for radial migration, and one could hypothesize that the observed delay for mutated neurons might stem from a reduced speed of migration and alteration of the rhythm of saltatory migration similar to what was observed in tangential migration.

Of interest, in a totally different system, the long distance migration of neural crest cells, mutation of the ciliary BBS (Bardet–Biedel Syndrome) genes similarly leads to a severe delay in migration [17], suggesting once again a similar function for the PC.

As a conclusion for this part, through the analysis of ciliary-ablated migrating neurons in different systems, the PC appears to be an essential regulator of neuronal migration, with the regulation of the rhythm of cyclic saltatory migration as the lowest common denominator.

## 4. The PC and the Intracellular Control of Migration

An important question is then to understand how the PC, a tiny organelle, often at a distance from the jumping nucleus, could intracellularly regulate the rhythm of the nucleus’ cyclic movement.

To understand this, it seems important to characterize the dynamic of the PC with respect to the two-stroke cycle of migration. The first clue concerning this arose from the study by Baudoin et al. [12], partially described above. The authors performed an EM analysis of medial ganglionic eminence migrating interneurons in culture. In approximately one third of the neurons with a large distance between the centrosome and the nucleus, i.e., probably starting or performing nucleokinesis, a short PC (500 nm) protruded from the mother centriole into the extracellular space. When the centrosome was closer to the nucleus, i.e., probably when the neuron is in pause, the mother centriole was sometimes associated with a vesicle, with an occasional short axoneme protruding, suggesting elongation of a PC in the ciliary vesicle (see Figure 1B above). The PC in migrating neurons thus seems to display cyclical behavior, alternating between an intracellular state during pauses and an extracellular state before or during nucleokinesis. The authors also clearly demonstrated that the centrosome, in addition to displaying PC assembly capacity, is directly associated with radially extended microtubules, clearly showing its double function as basal body and Microtubule Organizing Center (MTOC) in migrating neurons.

The parallel study of interneurons by Higginbotham et al. [11] also reported an extremely dynamic PC through live-imaging of Smoothened-TdTomato transfected neurons but did not study in detail either the orientation nor the cyclicity of its emergence. Another study also described a similar dynamic of the Arl13B-labeled PC in migrating cerebellar granule neurons [58], albeit without analysis of its potential cyclical emergence.

A detailed analysis of the PC intracellular localization was finally performed by Matsumoto et al. [13] by combining live-imaging of cultured postnatal rostral migratory stream neurons in Matrigel with EM analysis of adult rostral migratory stream. Three dimensional live-imaging of cultured neurons expressing Arl13B-Venus and Cent-GFP revealed that the PC cyclically emerges from the cell surface before nucleokinesis during swelling formation, whereas it appears mostly submerged intracellularly during pauses. By EM analysis of adult rostral migratory stream neurons, the authors confirmed the dual function of the centrosome as basal body and MTOC as in [12]. They also observed two states for the PC that they classified as non-extended and extended, similar to what was described in [12]. A mature PC was observed in 60% of the neuroblasts, with 80% of them classified as non-extended and 20% extended. The proportion of extended PC was increased during the swelling phase of formation, i.e., before nucleokinesis. Its length was, on average, 600–700 nm, independent of its extension state. Globally, the interpretation of these EM and live-imaging data was that the PC is submerged in the cytoplasm during pauses, that it extends from the cell surface when the swelling forms before nucleokinesis, moves forward, and finally relocates intracellularly during nucleokinesis and/or subsequent pauses (Figure 3A). Interestingly, the PC orientation was analyzed, showing that extended PCs were mainly oriented in the direction opposite to migration, which is not consistent with the role of the PC as a pointer towards the direction of migration as described in fibroblasts.

This cyclical emergence of the PC in migrating neurons thus appears to be present at least in embryonic interneurons [12] and rostral migratory stream neurons [13], suggesting a general behavior of the PC during neuronal migration. How this cyclical emergence is molecularly regulated is a fascinating question, which remains totally unexplored to date. In addition, to our knowledge, this phenomenon has never been reported for non-neuronal migrating cells. It is thus extremely tempting to correlate the PC cyclicity with the cyclic saltatory migration itself, and to imagine that the PC emergence could be a cause and/or a consequence of the cyclical movement of the saltatory nucleus. In this tantalizing scenario, the tiny PC could appear like a kind of piston for activation of the intracellular molecular motors. In addition, one could imagine that the cyclical PC emergence could allow the PC to periodically sense the extracellular environment.

In [14], we started to partly uncover how the PC could transduce extracellular cues. As the PC is a known rich cAMP source [23,24,25], we analyzed the cAMP dynamics in migrating rostral migratory stream neurons, by transfecting them with a cAMP-specific FRET biosensor (which, importantly, could not enter the PC but freely diffused in the cytoplasm). Unexpectedly, we discovered a transient cAMP hotspot subcellularly localized at the centrosome of migrating neurons before or during nucleokinesis and absent during pauses. This cAMP hotpot was PC-dependent since absent in ciliary mutants. The knockdown of the ciliary adenylate cyclase 3 also led to hotspot disappearance. This strongly suggested that the cAMP hotspot visible at the centrosome was produced by the ciliary adenylate cyclase 3 with the ciliary cAMP diffusing from the cilioplasm to the basal body/MTOC (Figure 3). We analyzed the migration of neurons with no hotspots, i.e., PC-ablated or adenylate cyclase 3 knocked-down neurons. As reported above for PC ablation, the speed of neurons was reduced with longer pauses and less frequent nucleokinesis. Moreover, mutated neurons also displayed defective centrosome dynamics. We then hypothesized that the centrosomal hotspot might be the activator of the cAMP-dependent Protein Kinase A (PKA). We showed that, indeed, PKA is enriched at the centrosome of rostral migratory-stream migrating neurons. We then used a dominant negative form of the PKA (dnPKA) [59], which prevents centrosomal anchoring of the endogenous PKA. Interestingly, the dnPKA-transfected neurons displayed the same migratory phenotype as the ciliary and adenylate cyclase 3 knocked-down mutants. In addition, this phenotype was not worsened in double mutants with adenylate cyclase 3 knock-down and dnPKA, strongly suggesting that centrosomal PKA is the downstream effector of the cAMP hotspot and hence the downstream effector of the PC. We finally showed that this regulation of neuronal migration can be generalized to different kinds of migrating neurons as we observed it in radially migrating neurons of the postnatal olfactory bulb but also in the embryonic cortex, as well as in adult rostral migratory stream neurons. As these neurons are also ciliated, one can suppose that, similar to what we analyzed in postnatal rostral migratory stream neurons, the PC and centrosome form a single cAMP unit where the PC is a cAMP producer through adenylate cyclase 3 activation and centrosomal PKA is the downstream effector. Of course, the downstream targets of the centrosomally activated PKA still have to be identified to understand how they can act on the neuronal cytoskeleton. It is to be noticed that the cyclicity of the hotspot, appears concomitant with the cyclic emergence of the PC (Figure 3). One can thus imagine that the emergence of the PC before nucleokinesis might expose a ciliary GPCR coupled to Gs to an extracellular cue, thus activating adenylate cyclase 3 and allowing the occurrence of the hotspot and transient cAMP action on the centrosomal PKA. Identification of the GPCR and cues involved is, of course, of paramount importance to completing the ciliary puzzle of neuronal cyclic saltatory migration.

This will allow us to fully understand how the cyclical behavior of the PC could be the origin of the cyclical movement of neurons and hence of the unicity of this type of cellular migration. This will also allow the discovery of a new functional modality for this ever-surprising little organelle.

## Figures and Tables

**Figure 1 cells-11-03384-f001:**
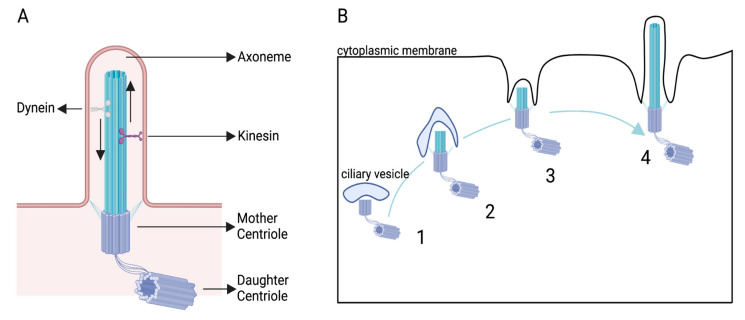
Ciliogenesis. (**A**) Scheme of the PC. (**B**) Model of ciliogenesis, B1, and formation of the ciliary vesicle adjacent to the mother centriole. B2, the PC starts to grow intracellularly. B3, the basal body moves within the cell to reach the membrane where the ciliary vesicle fuses with the cytoplasmic membrane. B4, the PC expands extracellularly.

**Figure 2 cells-11-03384-f002:**
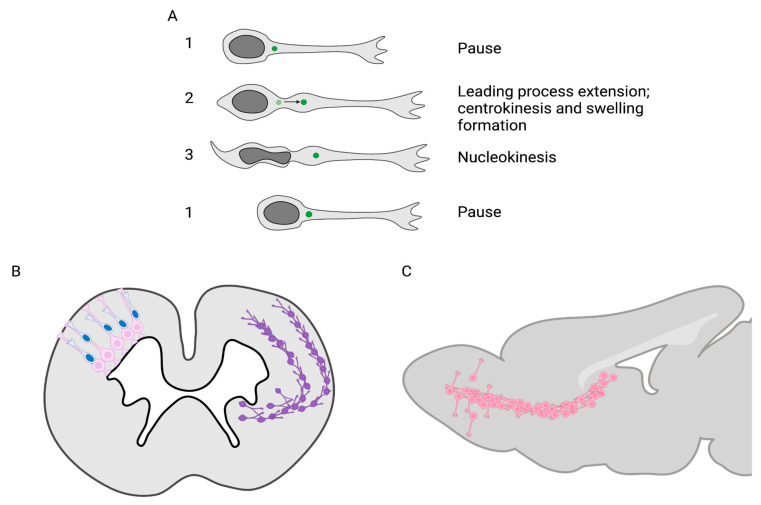
Neuronal migration. (**A**) Neurons perform cyclic saltatory migration. 1. The neuron is in a pausing phase. 2. The centrosome migrates in a forming swelling in the proximal leading process (centrokinesis). 3. The nucleus moves forward (nucleokinesis). 1. The neuron pauses again, and the cycle can start again (**B**) In the embryonic cortex, two types of migration occur: the radial glial-guided migration (shown in the left hemisphere) and the tangential migration (shown in the right hemisphere). Left: excitatory neurons are generated in the ventricular zone and migrate radially short distances along radial glia. Right: inhibitory interneurons are generated in the ganglionic eminences and migrate long distances tangentially in two different streams before re-orienting radially to the cortical plate. (**C**) In the postnatal and adult Rostral Migratory Stream, neuroblasts are generated in the ventricular/subventricular zone and migrate tangentially to the olfactory bulb, where they reorient and migrate radially to integrate the pre-existing network.

**Figure 3 cells-11-03384-f003:**
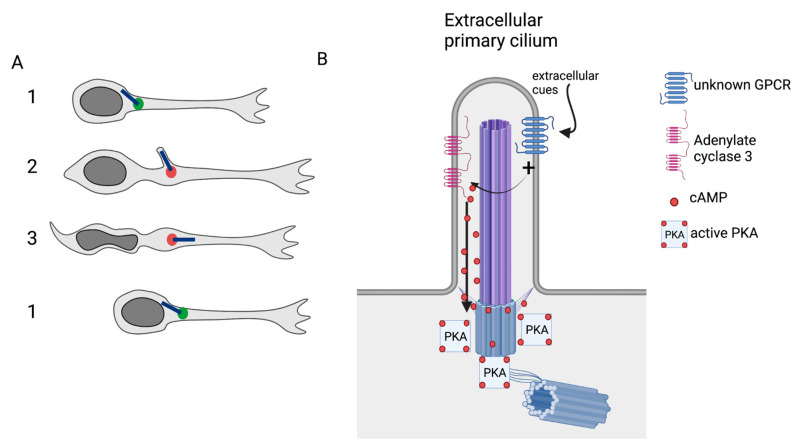
How does the PC control neuronal migration? (**A**) Dynamics of the PC and occurrence of the hotspot (red dot) during cyclic saltatory neuronal migration [13,14]. (**B**) While the PC is extracellularly exposed, an extracellular cue might activate a ciliary GPCR (G protein coupled receptor) coupled to Gs, which activates adenylate cyclase 3, leading to cAMP (cyclic adenosine monophosphate) production. The PC concentrates the cAMP that diffuses to the centrosome, where it activates PKA (protein kinase A), leading to phosphorylation of targets regulating neuronal migration.

**Table 1 cells-11-03384-t001:** Summary of the effects of ciliary mutations on neuronal migration.

References	Cell Type	Type of Migration	Ciliary Gene Depleted	Experiment to Study Migration	In Vivo, Ex Vivo, In Vitro	Cell Staining	Effects or Not
Higginbotham et al., 2012 [11]	Cortical Interneurons	Tangential migration	Arl13b	Live-imaging and fixed tissues	In vivo and ex vivo	Genetic labelling	Altered migration and directionality
Baudoin et al., 2012 [12]	Cortical Interneurons	Tangential migration	kif3a, IFT88	Live-imaging and fixed tissues	In vivo and ex vivo	Genetic labelling or acute brain slices electroporation	Altered migration and directionality
Matsumoto et al., 2019 [13]	RMS Neuroblast	Tangential migration	kif3a, IFT88	Live-imaging	In vitro	Nucleofection on V-SVZ explants	Altered migration
Stoufflet et al., 2020 [14]	RMS Neuroblast	Tangential migration	kif3a, Rpgrip1L	Live-imaging	Ex vivo	Postnatal electroporation	Altered migration
Higginbotham et al., 2012 [11]	Cortical projection neurons	Radial migration	Arl13b	Fixed tissues	In vivo	Genetic labelling	No effects
Guo et al., 2015 [10]	Cortical projection neurons	Radial migration	AHI1, ALMS1, BBS1, BBS4, BBS7, BBS9, BBS10, BBS11, BBS12, BUBR1, IFT80, KIF7, NPHP1, NPHP8, TCTN2, TMEM216, TUB	Fixed tissues	In vivo	In utero electroporation	Retarded migration
Pruski et al., 2019 [15]	Cortical projection neurons	Radial migration	IFT172	Fixed tissues	In vivo	In utero electroporation	Retarded migration
Park et al., 2018 [16]	Cortical projection neurons	Radial migration	ATG5	Fixed tissues	In vivo	In utero electroporation	Retarded migration
Tobin et al., 2008 [17]	Neural Crest cells	NA	BBS8	Fixed tissues	In vivo	In utero electroporation	Retarded migration

## Data Availability

Not applicable.

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
