# Peer review of "The Primary Cilium and Neuronal Migration"

_cells, 2022, doi:10.3390/cells11213384_

Round 1
Reviewer 1 Report
The narrow and informative review article by Isabelle Caillé and Julie Stoufflet summarizes the role of the primary cilium in neuronal migration of mice data, as far as this reviewer understand it. Some general explanations of the primary cilium, ciliopathies, neuronal cells, and the mentioned pathways and marker proteins (ace-tub, Arl13B, Kif3a, IFT88) should be added to the main text (including additional references) to be comprehensible to a broader readership. The included figures are nicely organized and comprehensive. However, this reviewer has included some recommendations for the authors below that should be addressed:
*Concerning the cilia length (line 35; please replace “,” with “.”), others report 2–10 μm in length and up to 200 µm (Nachury and Mick, Nat Rev Mol Cell Biol, 2019).
*line 39: it could be added, that the IFT machinery is coupled to the microtubule motor proteins kinesin-2 or dynein-2 to transport proteins along the axoneme in an anterograde or retrograde direction, as nicely illustrated in Fig. 1.
*line 40: the discovery of the primary cilium was in the 19th century; this could be added.
*line 105/6: “However, the molecular link between PC signaling and cytoskeleton remodeling remains mostly unstudied.” There are a lot of articles linking the role of the cytoskeleton (for example actin) to the primary cilium, also in neuronal cells, but probably not for the topic migration. This sentence should be reconsidered.
*In the part “The PC as a migration guide in non-neuronal cells”, please highlight, which studies used mouse embryonic fibroblasts or adult human fibroblasts. Are these all the studies on this topic or just a selection? This could be mentioned.
*line 128; please name some important genes.
*Most important: since there are few studies on this topic (PC in neuronal migration), perhaps a clear table would be advisable (including Ref, cell line/animal, study design/experimental setup, functions and molecular mechanism).
*It may also be possible to supplement the data with experiments using neuronal cell lines that have studied primary cilia/migration to better represent the data.
Since MDPI has no limitations for the reference list, additional references should be included. And for citations, the reference should be mentioned in the first sentence and also in the last. There are several paragraphs in the review that are not covered by references:
For example:
*line 50-53
*line 141-150
*line 157-180; refers only to Ref34? Is this whole topic only covered by two references?
*line 279-297; refers again only to ref 33
*ref 33 and 34 are often used throughout the manuscript; that are the only references dealing with that topic?
*line 321-323 refers to? Ref 33?
Figure legends:
*for MDPI journals, the abbreviations must be explained in the legend and also in the main text
Please add a Conclusion paragraph to your review article to summarize the key findings and future directions.
Author Response
We thank the referee for his/her positive analysis of our review and have considered his/her suggestions with great attention. Indeed, our review is meant to be narrow and informative, as mentioned by the referee
As suggested, we have added more references and additional info on the markers and pathways.
L35: concerning the cilia length, exactly as mentioned in the Nachury and Mick review, the PC length does not exceed 10µm (the 100µm length applies to the cilia of olfactory neurons which are not primary cilia).
L39: we have added the molecular motors to the text
L40: the discovery in the 19th century is added
L105: sentence reconsidered as suggested
In the part “the PC as a migration guide”, yes, we think that we have listed all studies which are relevant to the topic.
We are adding which fibroblasts are involved in the text.
L128: we have added some important genes
We have also added a table as suggested
To our knowledge there are no studies on cell lines.
Additional references have been added
*line 157-180; refers only to Ref34? Is this whole topic only covered by two references? yes it is indeed the case. As we more carefully explained in the text, the role of PC in neuronal migration has been covered by very few studies.
*line 279-297; refers again only to ref 33? Yes
*ref 33 and 34 are often used throughout the manuscript; that are the only references dealing with that topic? This is right
*line 321-323 refers to? Ref 33?: and 39, this has been added
Figure legends have been modified
The last paragraph is a conclusion
Reviewer 2 Report
The manuscript is interesting but needs to be significantly improved.
Major concern
- The paper is hard to read; it contains a lot of jargon and unnecessary acronyms
- it is not clear that the defects described in chapter 3 are a direct consequence of PC function. Can the phenotype be due to an indirect function of the PC, such as a partial change in cell phenotype? – please explain
- at what cell cycle stage in the neuron cell biology the mother centriole forms the cilium. Is it Before, during, or after the initiation of migration?
- The paper read like a Ph.D. thesis introduction and futter direction of a specific lab and not like a review.
Minor concerns:
Fig 1A – please indicate who the mother centriole is. The mother centriole should be placed below the cell membrane and not inside the cilium. The triplet of the centriole needs to be only ~2/3 of the length of the centriole; the rest is doublets.
- line 35: 0,5 should be 0.5
- what is known about the role of G-beta Gama in cilium activation
- Chapter 2 – are there ensamples of cells where the PC is not essential for cell migration?
- Chapter 3 – Please explain the difference between centrosome and cilium
- Fig 2A, B, C– what is the red dot in the cell? Please explain all colors in legend.
- please avoid using acronyms like IN, NK, CK, EM, dnPKA, and others.
- Line 223: what are P1 and P2?
- what is ATG5? Why is it important?
- Fig 3A1and 4: where is the centrosome? We need a more detailed explanation of what we see in the figure
- what is the localization pattern of AC3 in the cilium?
-“this phenotype was not worsened in double mutants, strongly 348 suggesting that indeed centrosomal PKA is the downstream effector of the cAMP hotspot 349 and hence the downstream effector of the PC.” Is it possible that one of the mutants had a maximal response, which cannot be additive?
Author Response
We thank the referee for his/her comments. We understand that the understanding of the topic might be difficult for somebody who is not familiar with neurons. We have tried to clarify the introduction to this extent.
Major concerns
We have removed as much jargons and acronyms as possible.
With all due respect, the defects reported in part III cannot be a change of cell phenotype: the cells are migrating neurons in all studies reported and remain migrating neurons both in terms of behavior and markers.
As far as the primary cilium is concerned, it has to be noticed that in nearly all articles described two different mutations were performed to induce the genetic ablation of the primary cilium. The similarities of induced phenotypes clearly show that they are consecutive to the primary cilium ablation rather than to a potential extraciliary function of the mutated protein.
As far as the cell cycle is concerned, neurons are postmitotic cells and hence not in the cycle. Concerning the migratory cycle, at which moment the PC forms is an essential one but not easy to answer technically. All the known elements (electron microscopic analysis mostly) are reported in Baudoin et al and Matsumoto et al as reported in detail in part 3.
We do not think that our review reads like a thesis introduction. As the referee 1 says it is a narrow and informative review on the role of the PC in neuronal migration, an important issue both in neuroscience and in the primary cilium field. The number of papers which really analyzed the PC’s role in neuronal migration is small and we tried to analyze them as thoroughly and extensively as possible. We do think it is a necessary review, as people mostly write that the PC is essential to neuronal migration without giving the real arguments, which are all clearly presented for the first time in our review.
Minor concerns
-Fig1A: the mother centriole is now indicated
-The change to 0.5 has been done
-we do not understand what the ref means by G-beta Gama, sorry
-Part 2: we are not aware of studies where the PC would not be essential for migration but a negative result is difficult to prove and even more difficult to publish
-Part 3: we think that the difference between centrosome and cilium is well described and represented in the figures
-Fig 2 is now clearer
-we removed the acronyms
-223: P1 and P2 are postnatal days, we changed that
-ATG5 is now explained
-Fig 3 and 4: now better explained
-for the two last points, with all due respect, we have to say that the answers to these points are present in the Stoufflet et al paper, which was already reviewed and accepted by others in a high profile journal
Round 2
Reviewer 1 Report
The review is strongly improved and the included table is a good overview. Thank you.
Author Response
Thank you for your suggestions.
Reviewer 2 Report
"The first hypothesis was that the PC was an evolutionary remnant organelle with no real function. The second one was that the PC was involved in sequestering the centriole to avoid mitosis entry. The third, and now proven to be true, hypothesis was that the PC is a key cellular sensory organelle." These hypotheses need references.
"Whereas the role of the PC in tangential migration was clearly demonstrated in 2012, it took surprisingly more time to prove its role in radial migration" These hypotheses need references.
Author Response
Done